# Impact of the Proportion of Foreign Players' Appearances on the Success of Football Clubs in Domestic Competitions and European Competitions in the Context of New Culture

**Michal Varmus \*, Milan Kubina and Roman Adámik**

Faculty of Management Science and Informatics, University of Zilina, Univerzitna 8215/1, 010 26 Zilina, Slovakia; milan.kubina@fri.uniza.sk (M.K.); roman.adamik@fri.uniza.sk (R.A.)

\* Correspondence: michal.varmus@fri.uniza.sk; Tel.: +421-41-513-4471

**Abstract:** The policy of using foreign players and their impact on club success is still relevant and can be viewed from multiple angles. This article focuses on the impact of the proportion of foreign players' appearances on the success of football clubs in their home leagues, as well as in European competitions. The research is focused on the most successful European clubs over the last ten years. A total of 19 European clubs were selected for research purposes, which resulted in 190 cases. Statistical tests, correlation, and regression analysis were used for their verification. The research shows that there is a dependence between the appearances by foreign players in matches and the success of clubs in domestic and European competitions. Furthermore, the trend of higher use of foreign players compared to domestic players was also proved. The presented research provides a different perspective on the issue, attributing greater importance to the appearances by players in matches compared to other studies, considering only the total number of foreign players.

**Keywords:** football; foreign player; appearance in the match; success; impact; culture

## 1. Introduction

When we look at the past, the original principle of building football clubs was based on a geographical principle. The clubs brought together players from the surrounding area and were therefore adapted to the cultural tradition. However, the development of scouting and the global market meant a turning point in professional sports. Most of the world's leading clubs have begun to recruit the best foreign players not only from neighboring countries but also from around the world [1]. This had an impact on the traditions and values that are an important part of the culture of a sports environment, because in sports organizations, traditional cultural values are deeply rooted. The dominant values represent the strongest standards that are maintained by the majority of the organization. Typically, the dominant values are part of the goals of the organization, and they underpin the philosophy and ideology of the organization [2].

These changes in the thinking of football clubs are closely related to globalization and thus bring a completely new perspective and thus a new culture to the football environment. Not only is globalization a significant part of modern football, but also football is a significant part of globalization; football has both reflected and advanced the globalization process in a variety of ways. On the other hand, globalization increases competition for the less attractive domestic competitions in smaller countries, which have to fight for fans with renowned overseas sports leagues [3,4].

Football clubs in England were formed based on religious missions (Aston Villa—founded 1874) or on factories (Arsenal—founded 1886) [5] attributes the embracing of football by the working class at

this time not only to its simplicity, but also to its ability to express a local sense of community, which had been lost in the movement from the rural environment to the amorphous mass of the city. This reflects the absolute absence of foreign players in English clubs at the time [5]. Later, in 2011/12 season of the Premier League (the top division of the English football league), English players accounted for just over a third of all appearances (37%), compared to just over two-thirds (69%) in the first season of the Premier League in 1992–1993 [6]. In the 2007/2008 season, the top ten highest goal scorers were already all foreign players, which means that foreign players brought quality to football in England [7]. Only 21 English players played in the 2014–2015 group stages of Champions League, despite there being four English teams (the highest number from country, together with Spain and Germany). The open market and presence of top foreign players has changed the culture of the English football league, which is now considered to be the best league in the world [6]. The expansion of Spanish sports clubs is also connected with foreigners. In Spain, modern sporting practices began to develop later than in other European countries—in the last quarter of the nineteenth century. The foundation of many Spanish clubs was closely associated with foreign professionals and specialists, who came to Spain to work in the industries before being part of the foundation of the clubs, which existed only as amateur clubs before this [8]. In Germany, the number of foreign players has grown over the years too. In the first German Bundesliga campaign (1963), there were just three foreign players. Forty years later, on the final day of the 2008 season's Bundesliga campaign, the number was above 150 [7].

In the context of these and other similar cultural changes in sport, research into the importance and impact of foreign players in the domestic leagues has become very relevant. According to Taylor [9], football migration has a long and complicated history, evidenced in accordance with the professionalization of football in Europe dating back in the early 1950s. This issue has been addressed by several authors. Poli [10] identified that after the increase in player migration flows in all European championships, the majority of players were from Eastern Europe and Latin America. Travlos, Dimitropoulos and Panagiotopoulos [11] dealt with the impact of foreign players on the domestic competition. Their research, however, was limited to the Greek league.

They researched a radical increase in foreign players in the Greek championship during the period of the study (2001–2013). They also studied the positive and significant statistical relationship between the investment in foreign talent and the position of the clubs in the championship and found that increased foreign player migration in football enhanced streams of revenues accrued from the media and sponsors allowing clubs to spend more on playing talent. Della Torre, Giangreco, Legeais, and Vakkayil [12] have been researching the impact of foreign players in the Italian league, with a particular emphasis placed on the differences in players' financial remuneration and the relationship between their current salary and current performance. They found evidence of complex discriminations regarding the influence of the player's origin on the performance–pay relationship, which give advantages to higher-performing domestic Italian players and lower-performing migrant players. In their research, Flores, Forrest, and Tena [13] evaluated the goal of Europe's strongest clubs, focusing more on the talent than the origin of a player, thus the contribution of the global market to diverse nationalities in the most ambitious teams. They also noted that the global market has contributed to a greater competitiveness in European football, but at the same time they admit a leader–follower relationship, such that strong clubs have first pick of the talent. Madichie [7] understood globalization in football as a tool that expands not only the sport but also the marketing opportunities and thus the potential income of clubs. However, his study only deals with the English Premier League. He admitted that globalization and the influx of foreign players into the Premier League has undoubtedly changed the face of English football. He also claimed that not only the English league, but all the major European Football Leagues, have become big businesses and deserve research attention like the more established corporations. Royuela and Gásquez [14] consider international migration as a very important phenomenon, which is nothing new in football. They also claimed that football clubs try to hire the best players, no matter where they come from, while football players aim to join the best clubs in order to enjoy better salaries and professional prospects, and that in a

globalized sport such as football, talent can be anywhere and this results in an international dimension, probably larger than in any other profession. In their research, they focused on the impact of foreign players on the results of more than 1000 clubs around the world. Their research showed that having more foreign players only has a positive effect for clubs in football confederations where a learning process can ultimately benefit home clubs. However, they found no dependence and concluded that more solvent clubs can choose better foreign players, because all clubs have the same possibilities for hiring better players. Bullough, Moore, Goldsmith, and Edmondson [6] claimed that modern football is a commercial product, therefore governors of the elite leagues are interested in creating the most commercially viable product to sell to broadcasters, sponsors etc., which creates an organizational ideology that is not necessarily aligned or mutually exclusive with indigenous player development, but rather player development regardless of nationality. In their research they focus on six elite European leagues (English, Spanish, German, Dutch, Italian, and French). They note that only the Spanish and Dutch leagues have created conditions for the use of domestic players. On the other hand, especially in English and Italian leagues, the foreign players spend most of their time on the pitch. For Hardman and Iorwerth [15], the weakening of national teams under the impact of a large number of foreign players is not necessarily true. They considered the rules of individual football federations as objective, and they attributed this effect to the officials from the domestic leagues who allow a large number of foreign players to play. At the same time, they claimed that developing opportunities for national talent involves more than simply limiting the number of foreign players in domestic leagues. Balsmeier, Frick, and Hickfang [16] evaluated the impact of foreign players on domestic players as positive. They perceived both the sporting and economic impact of foreign players as positive, as the foreign players can enhance not only the skills of domestic players, but also attract more fans to the matches of individual teams; however, they also note that the market for professional soccer players is highly competitive and highly specialized, and extraordinary individual performance does not necessarily translate into extraordinary team performance. Their research, however, was limited to the German league. Smith [17] focused only on the English league. He considered the participation of foreign players in English league clubs as a part of globalization and the global labor market. According to him, foreign players bring not only additional quality to the sport but also cultural principles which broaden the horizons of both domestic players and of the society as a whole, and the higher quality "education" and wider cross-cultural experiences gained by increasingly mobile overseas talent develop greater self-confidence. As already mentioned, several authors are concerned with the impact of foreign players, but their research is often limited either by focusing on a particular league or by the time intervals of their investigation. The available literature sources lack a comparison of the most successful clubs in Europe with an emphasis on the share of foreign players' appearances for these clubs. The aim of this article is to point out the impact of the number of appearances by foreign players on the success of clubs in Europe over the last ten years. The secondary objective is to point out the trends of the behavior of these clubs in their policy of engaging foreign players, thereby significantly affecting the traditional sports culture in the European region.

## 2. Materials and Methods

For the purposes of the research, data from available web portals for the last ten years were collected, relating specifically to the football seasons 2009/10 to 2018/19. The selection of the individual European clubs was chosen based on their success in international cup competitions for the period under review. We characterized success as first or second place in the UEFA Champions League and UEFA Europa League. The clubs that have achieved this success at least once have been surveyed. The individually surveyed clubs, as well as their placement, are shown in Table 1. The main indicators of the survey were the percentage of appearances by foreign and domestic players in matches, the number of matches in the season and the successes in the national and European leagues.

**Table 1.** Clubs and their positions in international cup competitions.

| Country | Club | Champions League Winner | Runner-up in the Champions League | UEFA Europa League Winner | Runner-up in the UEFA Europa League |
|---------|------|:-----------------------:|:---------------------------------:|:-------------------------:|:-----------------------------------:|
| | | Count | | | |
| | Manchester United | | 1 | 1 | |
| | Chelsea | 1 | | 2 | |
| **GBR** | Liverpool | 1 | 1 | | 1 |
| | Arsenal | | | | 2 |
| | Tottenham Hotspur | | 1 | | |
| | Real Madrid | 4 | | | |
| | Barcelona | 2 | | | |
| ESP | Atletico | | 2 | 3 | |
| | Bilbao | | | | 1 |
| | Sevilla | | | 3 | |
| GER | Bayern | 1 | 2 | | |
| | Dortmund | | 1 | | |
| ITA | Juventus | | 2 | | |
| | Inter | 1 | | | |
| | Porto | | | 1 | |
| POR | Braga | | | | 1 |
| | Benfica | | | | 2 |
| NED | Ajax | | | | 1 |
| FRA | Marseille | | | | 1 |

In order to evaluate the research, hypotheses were set which, in our opinion, best characterized the football environment in the researched issue and European cultural trends for sustainable success of clubs in international confrontation.

Within these hypotheses, we used the percentage of players' appearances in matches instead of the total number of players of the team. On the one hand, we believe that the quality or the success of the teams depends on the players who appear in the game and not on the players who are only in the squad. On the other hand, the total number of players would distort the results. The following hypotheses were formulated:

**Hypothesis 1 (H1).** *If there is a higher percentage of appearances by foreign players in national league matches, then the club is more successful in the national league.*

**Hypothesis 2 (H2).** *If there is a higher percentage of appearances by foreign players in European matches, then the club is more successful in the cup competitions (Champions League and UEFA Europa League).*

**Hypothesis 3 (H3).** *If the team plays more matches in the season, then more opportunities are given to domestic players.*

The individual hypotheses were verified by regression analysis. In addition, we used cluster analysis, the chi-squared test and correlation. All of the descriptive statistics were calculated by using the latest version of the Statistical Package for the Social Sciences (SPSS). In SPSS validation, these data were seen as panel data. Methods, such as induction, deduction, logic, synthesis, or comparison, were also used for the statistical processing of the data by means of said analyses, which provided the general complexity and consistency of the processed results.

## 3. Results

Within each team, we examined the number of domestic players, i.e., players from the country where the club operates and the number of players from abroad. As can be seen in Table 2, over the last ten years the proportion of foreign players in the observed clubs was about 55% of the total number of players, while the proportion of appearances in matches during individual seasons was on average more than 60%.

**Table 2.** Proportion of foreign players and players' appearances.

| | Season | | | | | | | | | |
| --- | --- | --- | --- | --- | --- | --- | --- | --- | --- | --- |
| | **2009/10** | **2010/11** | **2011/12** | **2012/13** | **2013/14** | **2014/15** | **2015/16** | **2016/17** | **2017/18** | **2018/19** |
| Proportion of foreign players (average) | 57.25% | 54.64% | 53.89% | 53.56% | 56.37% | 55.98% | 58.71% | 57.37% | 57.71% | 57.18% |
| Proportion of foreign players' appearances (average) | 60.44% | 60.26% | 57.06% | 55.85% | 60.37% | 60.13% | 62.17% | 61.9% | 60.72% | 62.09% |

For better clarity, we can put the individual teams into groups according to the country in which they operate. Figure 1 shows the development of the number of foreign players at clubs over time in relation to the total number of players at clubs.

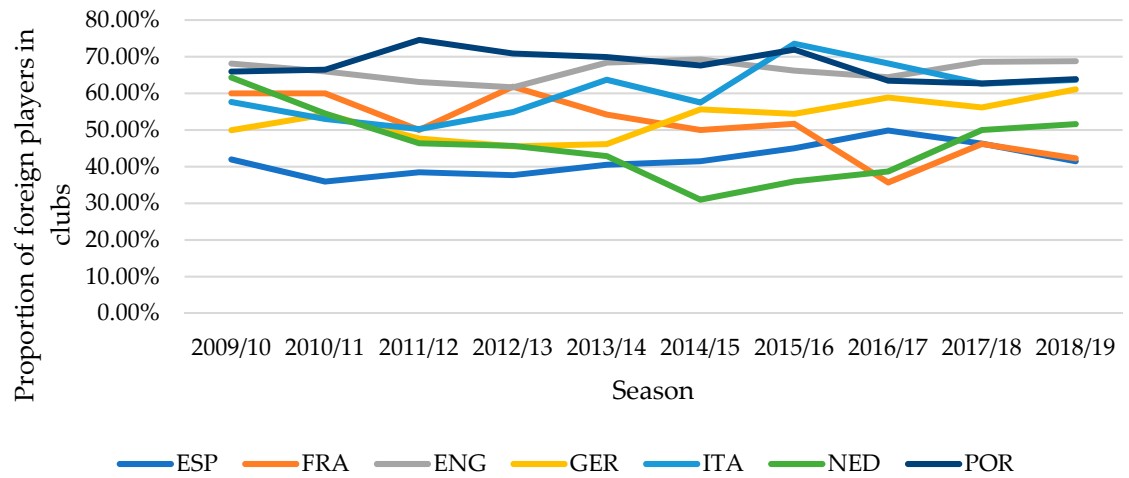

**Figure 1.** Development of the number of foreign players at clubs.

Table 3 below shows that the highest number of average appearances by foreign players in matches was in English teams and the smallest number was in Spanish teams.

**Table 3.** Clubs and their position in international cup competitions.

| | Country | | | | | | |
| --- | --- | --- | --- | --- | --- | --- | --- |
| | **ESP** | **FRA** | **ENG** | **GER** | **ITA** | **NED** | **POR** |
| Proportion of foreign players (average) | 41.8% | 51.19% | 66.45% | 52.98% | 60.5% | 46.11% | 67.73% |
| Proportion of appearances by foreign players (average) | 46.6% | 50.23% | 71.81% | 53.28% | 65.5% | 48.86% | 70.98% |

The trend by season is highlighted in the Figure 2. It can be seen that the changes were more or less noticed in all teams.

Since we assumed that teams with more foreign players are also more successful in national leagues, we set the following hypothesis H1.

H1 If there is a higher percentage of appearances by foreign players in national league matches, then the club is more successful in the national league.

Table 4 summarizes the results of the teams surveyed for the reference period 2009/10 to 2018/19.

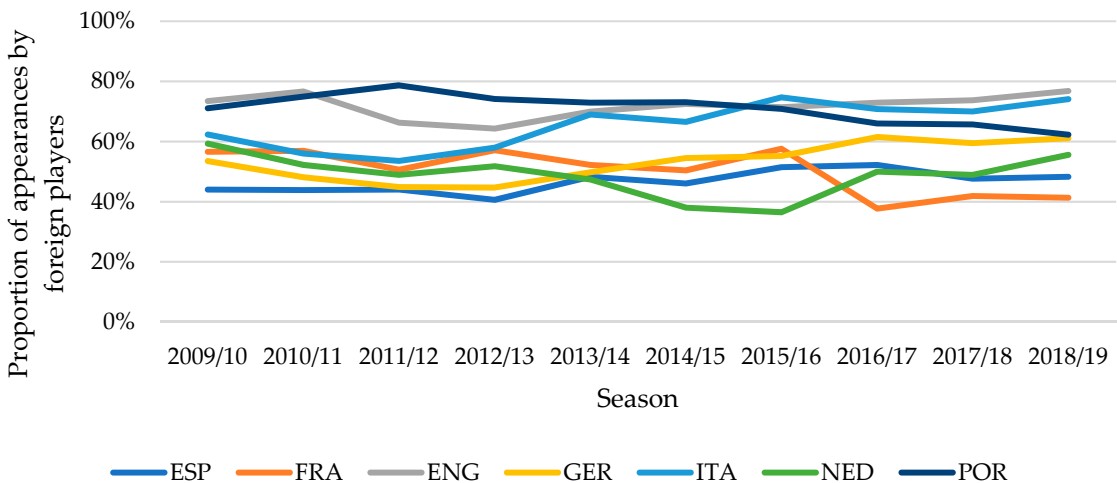

**Figure 2.** Trend of the teams in average appearances by foreign players by season.

**Table 4.** Results of the teams 2009/10 to 2018/19.

| | | National League Position for the Period 2009/10–2018/19 | | | | | | | | | | | | |
| | | 1 | 2 | 3 | 4 | 5 | 6 | 7 | 8 | 9 | 10 | 12 | 13 | 16 |
| | | Count | | | | | | | | | | | | |
| | **Ajax** | **5** | **5** | **0** | **0** | **0** | **0** | **0** | **0** | **0** | **0** | **0** | 0 | 0 |
| **Club** | Arsenal | 0 | 1 | 3 | 3 | 2 | 1 | 0 | 0 | 0 | 0 | 0 | 0 | 0 |
| | Atletico | 1 | 2 | 4 | 0 | 1 | 0 | 1 | 0 | 1 | 0 | 0 | 0 | 0 |
| | Barcelona | 7 | 3 | 0 | 0 | 0 | 0 | 0 | 0 | 0 | 0 | 0 | 0 | 0 |
| | Bayern | 8 | 1 | 1 | 0 | 0 | 0 | 0 | 0 | 0 | 0 | 0 | 0 | 0 |
| | Benfica | 6 | 4 | 0 | 0 | 0 | 0 | 0 | 0 | 0 | 0 | 0 | 0 | 0 |
| | Bilbao | 0 | 0 | 0 | 1 | 1 | 1 | 2 | 2 | 0 | 1 | 1 | 0 | 1 |
| | Braga | 0 | 1 | 1 | 6 | 1 | 0 | 0 | 0 | 1 | 0 | 0 | 0 | 0 |
| | Chelsea | 3 | 1 | 3 | 0 | 1 | 1 | 0 | 0 | 0 | 1 | 0 | 0 | 0 |
| | Dortmund | 2 | 4 | 1 | 1 | 1 | 0 | 1 | 0 | 0 | 0 | 0 | 0 | 0 |
| | Inter | 1 | 1 | 0 | 3 | 1 | 1 | 1 | 1 | 1 | 0 | 0 | 0 | 0 |
| | Juventus | 7 | 1 | 0 | 0 | 0 | 0 | 2 | 0 | 0 | 0 | 0 | 0 | 0 |
| | Liverpool | 0 | 2 | 0 | 2 | 0 | 2 | 2 | 2 | 0 | 0 | 0 | 0 | 0 |
| | Manchester United | 2 | 3 | 0 | 1 | 1 | 2 | 1 | 0 | 0 | 0 | 0 | 0 | 0 |
| | Marseille | 1 | 2 | 0 | 2 | 2 | 1 | 0 | 0 | 0 | 1 | 0 | 1 | 0 |
| | Porto | 4 | 3 | 3 | 0 | 0 | 0 | 0 | 0 | 0 | 0 | 0 | 0 | 0 |
| | Real Madrid | 2 | 5 | 3 | 0 | 0 | 0 | 0 | 0 | 0 | 0 | 0 | 0 | 0 |
| | Sevilla | 0 | 0 | 0 | 2 | 2 | 1 | 3 | 0 | 2 | 0 | 0 | 0 | 0 |
| | Tottenham Hotspur | 0 | 1 | 2 | 3 | 3 | 1 | 0 | 0 | 0 | 0 | 0 | 0 | 0 |

Table 5 shows the position of each team in the national league for the average reporting period. However, as mentioned above, a more important factor than the total number of foreign players is the percentage of foreign players' appearances, as it better reflects the involvement of foreign players at the club. In teams that won national leagues, the average appearance of foreign players in matches was more than 50%. Extremes also occur when, for example, in the Italian league, the number of appearances by foreign players was as much as 92% for the Inter Milan team, which finished in fifth place during the 2013/14 season.

**Table 5.** Position of each team in the national league and average percentage of appearances by foreign players.

| | | Country (Average Percentage of Appearances by Foreign Players) | | | | | | |
|---|---|---|---|---|---|---|---|---|
| | Ranking | ESP | FRA | ENG | GER | ITA | NED | POR |
| National League | 1 | 54.57% | 56.60% | 75.92% | 54.86% | 57.45% | 51.18% | 73.90% |
| | 2 | 57.08% | 57.00% | 68.65% | 48.70% | 67.40% | 46.54% | 79.38% |
| | 3 | 54.49% | | 77.30% | 49.70% | | | 70.70% |
| | 4 | 44.70% | 46.15% | 70.39% | 61.80% | 79.60% | | 59.10% |
| | 5 | 53.78% | 39.50% | 75.66% | 57.90% | 92.10% | | 59.50% |
| | 6 | 29.85% | 52.20% | 67.97% | | 81.40% | | |
| | 7 | 41.87% | | 65.47% | 54.30% | 46.87% | | |
| | 8 | 4.20% | | 60.10% | | 81.70% | | |
| | 9 | 49.97% | | | | 81.00% | | 58.40% |
| | 10 | 6.00% | 50.60% | 87.70% | | | | |
| | 12 | 6.50% | | | | | | |
| | 13 | | 57.60% | | | | | |
| | 16 | 4.00% | | | | | | |

A regression analysis was used to verify the hypothesis. The result is in the following Table 6.

**Table 6.** Model 1 Pooled OLS, using 190 observations including 19 cross-sectional units. Time-series length = 10. Dependent variable: National League.

| | Coefficient | Std. Error | t-Ratio | *p*-Value |
|---|---|---|---|---|
| foreign players | 0.0506361 | 0.00359279 | 14.0938 | <0.0001 |
| R-squared | 0.512427 | | Adjusted R-squared | 0.512427 |
| F(1, 189) | 198.6346 | | *p*-value(F) | $2.67 \times 10^{-31}$ |
| rho | 0.100693 | | Durbin–Watson | 1.661645 |

Only 51 % of the variability is explained by the model. On the other hand we used the chi-squared test to confirm the correlation between the percentage of appearances by foreign players in matches during the season and the success of the team in their national league. The test result is as follows:

Corr (league, proportion_guests) = −0.26340024
Under the null hypothesis of no correlation:
$t(188) = -3.74377$, with a two-tailed *p*-value 0.0002

To better demonstrate the ratio of the number of foreign and domestic players, a cluster analysis of the K-means cluster was also performed (Figure 3).

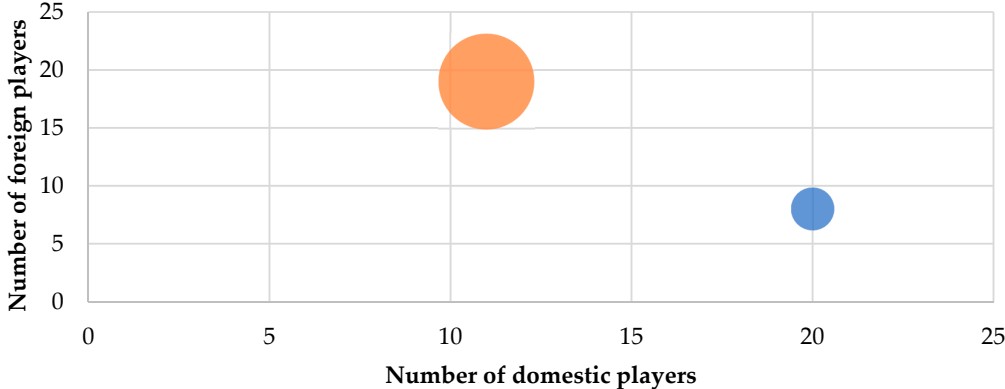

**Figure 3.** K-means cluster; number of foreign players vs. number of domestic players.

As can be seen, based on cluster analysis, out of 190 cases, at most (158) the number of foreign players is higher by almost half than the number of domestic players.

Even though chi-squared test confirmed the correlation between the percentage of appearances by foreign players in matches during the season and the success of the team in their national league, the model explained only 51 % of the variability. From our point of view the model is not statistically significant so the hypothesis H1 is rejected.

H2 If there is a higher percentage of appearances by foreign players in the matches, then the club is more successful in the cup competitions (Champions League and UEFA Europa League).

As in the previous case, we assumed that teams that have several foreign players are also more successful in international cup competitions. The hypothesis verification was divided into two parts, namely for the Champions League and separately for the UEFA Europa League.

Table 7 shows the success of the clubs of the specific countries in the Champions League and the average number of appearances by foreign players in the period under review.

**Table 7.** Position of teams in the UEFA Champions League and average percentage of appearances by foreign players.

| | | Country (Average Number of Appearances by Foreign Players) | | | | | | |
|---|---|---|---|---|---|---|---|---|
| | **Ranking** | **ESP** | **FRA** | **ENG** | **GER** | **ITA** | **NED** | **POR** |
| **UEFA** | 1 | 55.45% | | 72.40% | 54.40% | 84.90% | | |
| **Champions** | 2 | 56.25% | | 69.30% | 42.43% | 60.40% | | |
| **League** | 3–4 | 56.24% | | 79.40% | 57.18% | | 55.60% | |
| | 5–8 | 56.23% | 50.60% | 74.18% | 55.83% | 64.28% | | 80.04% |
| | 8–16 | 56.58% | 56.90% | 78.83% | 54.90% | 60.00% | | 72.28% |
| | Group | 44.08% | 54.40% | 67.82% | 54.20% | 54.93% | 46.71% | 74.29% |
| | Did not play | 33.76% | 47.67% | 66.82% | 52.90% | 69.16% | 59.30% | 62.48% |

A regression analysis was used to verify the hypothesis. The result is shown in Table 8.

**Table 8.** Model 2 Pooled OLS, using 190 observations, including 19 cross-sectional units. Time-series length = 10. Dependent variable: Champions League.

| | **Coefficient** | **Std. Error** | **t-Ratio** | ***p*-Value** |
|---|---|---|---|---|
| Foreign players | 0.0447922 | 0.00275165 | 16.2783 | <0.0001 |
| R-squared | 0.583686 | | Adjusted R-squared | 0.583686 |
| F(1, 189) | 264.9837 | | *p*-value(F) | $8.20 \times 10^{-38}$ |
| rho | −0.113835 | | Durbin–Watson | 1.996965 |

Only 58% of the variability is explained by the model. When we used the chi-squared test, there was a confirmed correlation between the percentage of appearances by foreign players in matches during the season and the success of the team in the Champions League. It can therefore be stated that the dependence exists. The test result was as follows:

Corr (Champions_League, proportion_guests) = 0.21047424
Under the null hypothesis of no correlation:
t(188) = 2.952, with a two-tailed *p*-value 0.0036

As can be seen in Figure 4 below, the cluster analysis demonstrates the relationship between success in the Champions League and success in the national league.

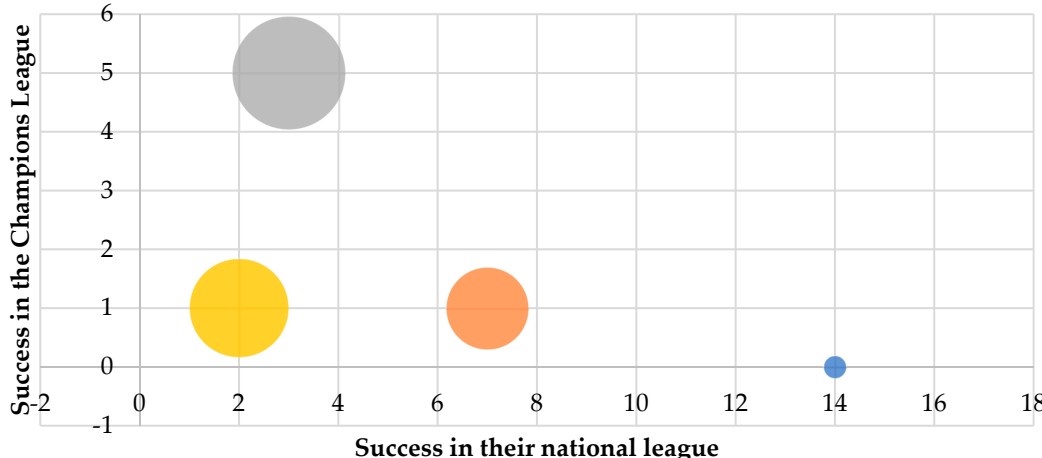

**Figure 4.** K-means cluster; success in the Champions League vs. success in the National League.

Table 9 shows the success of the clubs of the countries in the UEFA Europa League and the average number of appearances by foreign players in the period under review.

**Table 9.** Position of teams in the UEFA Europa League and average percentage of appearances by foreign players.

| | | Country (Average Number of Foreign Players) | | | | | | |
|---|---|---|---|---|---|---|---|---|
| | **Ranking** | **ESP** | **FRA** | **ENG** | **GER** | **ITA** | **NED** | **POR** |
| **UEFA** | 1 | 63.95% | | 76.37% | | | | 70.90% |
| **Europa** | 2 | 6.00% | 41.90% | 74.80% | | | 50.00% | 82.03% |
| **League** | 3–4 | | | | | 45.90% | | 78.20% |
| | 5–8 | 5.20% | | | 47.00% | | | 69.25% |
| | 8–16 | 43.03% | 56.60% | 61.27% | 61.80% | 72.60% | 45.10% | |
| | Group | 29.17% | 52.00% | 62.18% | 53.80% | 50.40% | 48.80% | 64.47% |
| | Did not play | 50.82% | 49.56% | 74.50% | 53.11% | 66.69% | 55.60% | 72.41% |

A regression analysis was used to verify the hypothesis. The result is in Table 10.

**Table 10.** Model 3 Pooled OLS, using 190 observations including 19 cross-sectional units. Time-series length = 10. Dependent variable: UEFA Europa League.

| | Coefficient | Std. Error | t-Ratio | *p*-Value |
|---|---|---|---|---|
| foreign players | 0.0249569 | 0.00305134 | 8.1790 | <0.0001 |
| R-squared | 0.261418 | | Adjusted R-squared | 0.261418 |
| F(1, 189) | 66.89583 | | *p*-value(F) | $4.09 \times 10^{-14}$ |
| rho | −0.052807 | | Durbin–Watson | 1.898875 |

Only 26% of the variability is explained by the model. When we used the chi-squared test there was a confirmed correlation between the percentage of appearances by foreign players in matches during the season and the success of the team in UEFA Europa League. It can therefore be stated that the dependence exists. The test result is as follows:

$$\text{Corr (UEFA\_Europa League, proportion\_guests)} = -0.21953760$$
$$\text{Under the null hypothesis of no correlation:}$$
$$t(188) = -3.08542, \text{ with a two-tailed } p\text{-value } 0.0023$$

In this case we also used cluster analysis. In Figure 5 below is demonstrated the relationship between success in the UEFA Europa League and success in the national league.

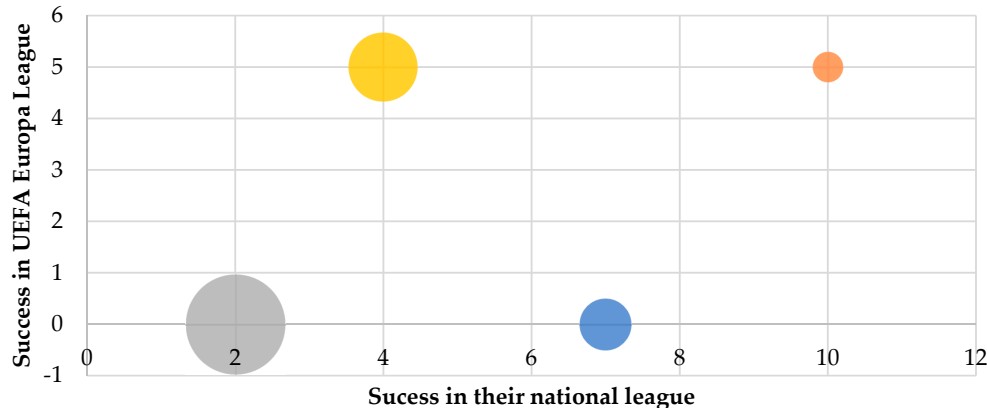

**Figure 5.** K-means cluster; success in the UEFA Europa League vs. success in the national league.

Chi-squared test confirmed the correlation between percentage of appearances by foreign players in matches during the season and the success of the team in the Champions League and UEFA Europa League. Model explained low percentage of the variability (Champions League 51%, UEFA Europa League 26%). From our point of view the model is not statistically significant so the hypothesis H2 is rejected.

H3 If the team plays more matches in the season, then more opportunities are given to domestic players.

We assumed that the more matches that take place in the season, the more players, in this case the domestic players, get the chance to play. A regression analysis was used to verify the hypothesis. The result is in Table 11.

**Table 11.** Model 4: Pooled OLS, using 190 observations including 19 cross-sectional units. Time-series length = 10. Dependent variable: proportion_domestic.

|  | Coefficient | Std. Error | t-Ratio | *p*-Value |
|---|---|---|---|---|
| number_matches | 0.735526 | 0.0270254 | 27.2161 | <0.0001 |
| R-squared | 0.796712 |  | Adjusted R-squared | 0.796712 |
| F(1, 189) | 740.7162 |  | *p*-value(F) | $2.68 \times 10^{-67}$ |
| rho | 0.050769 |  | Durbin–Watson | 1.700662 |

Almost 80 percent of the variability is explained by the model, the coefficient of determination is statistically significant, so the model is also significant. Hypothesis H3 is accepted. To better understand the situation, a cluster analysis of the K-means cluster was also developed. A graphical representation of the result is shown in the following Figure 6.

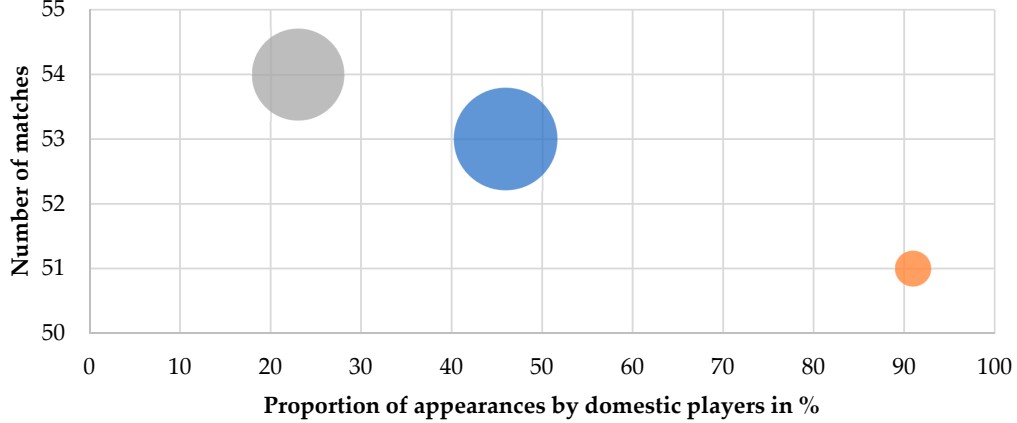

**Figure 6.** K-means cluster Number of matches vs. Proportion of appearances by domestic players.

The cluster with the most cases, namely the blue cluster containing 99 cases, indicates that with 53 matches there is a 49% proportion of appearances for domestic players. The largest proportion of domestic players' appearances is 91% corresponding with 51 matches as shown by the orange cluster with 12 cases.

## 4. Discussion

In professional circles as well as the lay public there are a lot of disputes as to which football league in Europe is the best one and which football school is better. Is it the English or the Spanish one? Are foreign players crucial for clubs? It is very difficult to find objective criteria, but despite this, one of them is the success in the European leagues. That is why this criterion was essential for the selection of the clubs examined. Although this article does not respond to everything, we have focused on the meaning, or more precisely the impact of foreign players on the success of clubs. We were therefore interested in those clubs that have been successful in the Champions League and the UEFA Europa League over the last ten years. We understand the importance of foreign players, on several levels. The number of foreign players in a club can signal an understanding of the national and club identity in Europe and have a significance for the sustainability of each club's success.

The open market and presence of top foreign players has changed the culture of football in Europe. In England, clubs were formed on community, based on an absence of foreign players [7]. Over the years, English clubs have changed fundamentally. In the 2011/12 season of the Premier League (the top division of the English football league), English players accounted for just over a third of all appearances (37%), compared to just over two-thirds (69%) in the first season of the Premier League in 1992–1993 [8]. In Germany, the number of foreign players has grown over the years too. In the first German Bundesliga campaign (1963) there were just three foreign players. Forty years later, on the final day of the 2008 season's Bundesliga campaign, the number was above 150 [6]. The foundation of Spanish clubs was closely associated with foreign professionals and specialists who came to Spain to work in the industries before being involved in the setting up of the clubs, which existed only as amateur clubs before this [8].

> If there is a higher percentage of appearances by foreign players in national league matches, then the club is more successful in the national league.

This is a result that speaks of the relatively important role of foreign players in the clubs that they play for in national leagues. The percentage of foreign players' appearances has an important position mainly because of increasingly strong voices and the implementation of quotas for foreign players in individual teams. Former FIFA President Sepp Blatter [16,18] was a major supporter of such quotas. His view had many supporters, but also critics such as Hardman and Iorwerth [11].

However, there may be a situation in which the clubs meet the quota in the line-up, but in matches, domestic players are given less opportunities than foreign players. As can be seen in the results of the research, English and Portuguese clubs have the largest number of foreign players. As for the English clubs, there is an interesting phenomenon, namely a significant proportion of foreign players' appearances in matches, with an average of 71.81%. It would be interesting to compare all of the clubs in the national leagues, although we do not assume that this would have any significant impact on the verification of the hypothesis.

Although the hypothesis was rejected, there was a proven dependence between the number of foreign player starts and success in the national league.

The reason may be that only the most successful teams in the league have been watched and it is therefore assumed that they can have more foreign players than clubs that have not been so successful. Other factors that may have an impact are not excluded.

If we look at the time horizon within the set horizon, we can see that the proportion of foreign players to domestic players had an alternating tendency. However, it has had a significant impact on

the team's success over the past ten years. It is also worth noting that the situation was also accepted by club fans and that club fans have no problem identifying with players from other countries.

> If there is a higher percentage of appearances by foreign players in European matches, then the club is more successful is in the cup competitions. (Champions League and UEFA Europa League)

As mentioned, this hypothesis was verified in two steps, separately for the Champions League and separately for the UEFA Europa League. As could be seen in the results, the hypothesis was not confirmed, although the dependence between the number of foreign player starts and the success in the cup competitions was proved. There may be a number of reasons for rejecting the hypothesis, but the most obvious seems to be the possibility of other factors affecting the success in these competitions.

The Spanish clubs in particular, which on average achieve a 55% proportion of foreign players' appearances and are among the most successful clubs in European competitions over the last ten years, are interesting in these results. Virtually all of the successful clubs had an average percentage of more than 50%. Of course, there were extremes, but they were more or less unique, or it was a one-off success.

> If the team plays more matches in the season, then more opportunities are given domestic players.

With an increasing number of matches, the logical assumption is that a wider range of players will be used, thus including players who receive fewer opportunities. Based on the hypothesis formulation, a regression analysis was used to confirm our hypothesis. This hypothesis confirms the phenomenon that the more matches are played, the more players get the opportunity from the bench. If we look at the dependence between participation by a higher proportion of foreigners and the success of clubs in European competitions as was mentioned in this paper, this may be related to the importance of matches, especially when we are talking about the play-off part of the competition.

## 5. Conclusions

The aim of this article is to point out the impact of foreign players on the sustainability of the success of teams at an international as well as national level. The results of the research show that the teams surveyed have a relatively significant number of foreign players and that these players are given more opportunities in matches than the domestic players. This suggests a significant shift in club thinking and understanding of club identity. It is well-known that the individual clubs enjoy significant attendance during their matches, so there is a situation where the traditional cultural and national identity from the viewpoint of clubs, and also of fans, is behind the club identity. Other research results show that the number of appearances by foreign players has an impact on the success of the clubs surveyed in their national leagues, but especially in international competitions. Thus, in the international competitions, the national aspect is significantly replaced by the club aspect.

The question arises as to why this is so. Why do clubs prefer to invest in buying foreign players? Is it cheaper than training their own players? Especially when we see a strong discrepancy between Spanish and English clubs in the research results.

Of course, this research also has its limits, as it does not include those clubs that have not been particularly successful in international competitions over the past decade, and the research has not included any results of successful clubs from the past. All this can be an interesting subject for further investigation, but in any case, it can be argued that foreign players are essential for club success in today's competitive environment.

**Author Contributions:** M.V. and M.K. coordinated the research, R.A. collected data, M.V. and M.K. analyzed the data, M.V. and R.A. wrote the paper, all authors built the conceptual framework, and all authors contributed to the manuscript preparation. All authors have read and agreed to the published version of the manuscript.

**Funding:** The paper was funding by the project VEGA 1/0617/16—Diagnosis of Specifics and Determinants in Strategic Management of Sporting Organizations.

**Acknowledgments:** Many thanks to all reviewers for their opinions and suggestions to make this paper better.

**Conflicts of Interest:** The authors declare no conflict of interest.

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
