# Peer review of "Impact of the Proportion of Foreign Players’ Appearances on the Success of Football Clubs in Domestic Competitions and European Competitions in the Context of New Culture"

_sustainability, doi:10.3390/su12010264_

Round 1

Reviewer 1 Report

This work does not have the parts that must appear in a scientific article.
1. The method should specify: a) participants, b) data analysis, c) procedures, among others.
2. Language is poor, must be improved.
3. The discussion should be much more elaborate.

Reviewer 2 Report

The paper under title "Impact of the proportion of foreign players’ appearances on the success of football clubs in domestic competitions and European competitions in the context of new culture" deals with the topic of football player migration in European football. Eventhought the paper may have interest for the journal's readers but it has serious issues relating to theoretical background, hypothesis formation and reasearch design that make it unsuitable for publication in its current form. I urge the authors to consider the following comments and improve the paper for potential future endeavour.

1) The paper requires an extensive editing by a native english speaker, because there are many syntax and grammatical mistakes throughout the paper.

2)The title says about a "new culture" but this is not substantiated in the paper. Authors must be more specific and explain in more detail about the new culture.

3)The first paragraph of the introduction is irrelevant to the topic of the paper. It confuses the reader and needs further analysis by the authors in order to help the readers to understand waht authors are discussing.

4) The introduction does not support a clear motivation and contribution to existing literature.

5)On the second section authors state that they collected by the first and second place of UEFA's champions league and Europa league. My question is why not extending the sample to the top-4 or top-8 clubs in these competitions since top-4 teams are also big clubs with significant investment in player contracts from abroad. Focusing only on the finalists may create a sample selection bias affecting statistical inference.

6) The research hypotheses need to be derived and supported by previous evidence and studies on the topic. As they are stated now they more look like with research assertions (not even questions) which are not based on specific reasoning.

7) The empirical analysis is very basic using coorrelations for H1 and H2 but a regression analysis is performed for H3. Why this research design is selected and why authors have not used regression analysis for the second and first hypotheses?

8) Also the statement on H3 provides the opposite argument compared to waht previously has been stated in the other hypotheses. 

9) The reference list requires several corrections since there are pages missing in some citations.  

Reviewer 3 Report

Dear authors,

I was pleased to read this article, which attempts to develop an international and longitudinal analysis of a very complex theme and issue in world football. Your various graphs and tables were very informative, and were on the whole very clear and comprehensible. 

In order to improve the depth and extend of the arguments made, I suggest the following:

- The introduction could offer more historical and regional examples of sporting traditions, especially in football. This would offer a clear link to the very long second paragraph. More flow could be established with the short sentences offered here. 

A more extensive literature review that provides more details on the studies and how they were conducted. You could also consider the other works published on the theme of football by these and other authors. Some key texts are missing here, e.g. work by Richard Giulianotti on globalisation. The methodology is generally sound, but could be supported by citations to literature on statistics. The results are clear apart from the coloured circles, which required a key for explanations. The discussion could feed back to the literature on football, and the critical points on FIFA required more explanation (e.g. on Blatter). In fact, the role of such governing bodies might need more consideration in light of its influence over national leagues, the Champions League, etc.  The conclusions could consider limitations of the current project and streams for further research.